# Methylation of histone H3K23 blocks DNA damage in pericentric heterochromatin during meiosis

Romeo Papazyan[1,2‡], Ekaterina Voronina[3,4†¶], Jessica R Chapman[5†#], Teresa R Luperchio[2,6], Tonya M Gilbert[1,2], Elizabeth Meier[1,2], Samuel G Mackintosh[7], Jeffrey Shabanowitz[5], Alan J Tackett[7], Karen L Reddy[2,6], Robert S Coyne[8], Donald F Hunt[5,9], Yifan Liu[10], Sean D Taverna[1,2]*

[1]Department of Pharmacology and Molecular Sciences, The Johns Hopkins University School of Medicine, Baltimore, United States; [2]Center for Epigenetics, The Johns Hopkins University School of Medicine, Baltimore, United States; [3]Department of Molecular Biology and Genetics, Howard Hughes Medical Institute, The Johns Hopkins University School of Medicine, Balitmore, United States; [4]Center for Cell Dynamics, The Johns Hopkins University School of Medicine, Baltimore, United States; [5]Department of Chemistry, University of Virginia, Charlottesville, United States; [6]Department of Biological Chemistry, The Johns Hopkins University School of Medicine, Baltimore, United States; [7]Department of Biochemistry and Molecular Biology, University of Arkansas for Medical Sciences, Little Rock, United States; [8]Department of Genomic Medicine, J. Craig Venter Institute, Rockville, United States; [9]Department of Pathology, University of Virginia, Charlottesville, United States; [10]Department of Pathology, University of Michigan Medical School, Ann Arbor, United States

*For correspondence: staverna@jhmi.edu

†These authors contributed equally to this work

Present address: ‡Division of Endocrinology, Diabetes, & Metabolism, Department of Medicine, Perelman School of Medicine at the University of Pennsylvania, Philadelphia, United States; ¶Division of Biological Sciences, University of Montana, Missoula, United States; #Proteomics Resource Center, New York University Langone Medical Center, New York, United States

Competing interests: The authors declare that no competing interests exist.

**Abstract** Despite the well-established role of heterochromatin in protecting chromosomal integrity during meiosis and mitosis, the contribution and extent of heterochromatic histone posttranslational modifications (PTMs) remain poorly defined. Here, we gained novel functional insight about heterochromatic PTMs by analyzing histone H3 purified from the heterochromatic germline micronucleus of the model organism *Tetrahymena thermophila*. Mass spectrometric sequencing of micronuclear H3 identified H3K23 trimethylation (H3K23me3), a previously uncharacterized PTM. H3K23me3 became particularly enriched during meiotic leptotene and zygotene in germline chromatin of *Tetrahymena* and *C. elegans*. Loss of H3K23me3 in *Tetrahymena* through deletion of the methyltransferase Ezl3p caused mislocalization of meiosis-induced DNA double-strand breaks (DSBs) to heterochromatin, and a decrease in progeny viability. These results show that an evolutionarily conserved developmental pathway regulates H3K23me3 during meiosis, and our studies in *Tetrahymena* suggest this pathway may function to protect heterochromatin from DSBs.

## Introduction

Eukaryotic genomes are organized around histone and non-histone proteins into at least two major functionally distinct states of chromatin that can be epigenetically inherited: heterochromatin, which is highly compacted and transcriptionally repressive; and euchromatin, which is relatively open and transcriptionally permissive. Heterochromatin, in addition to helping control transcription, plays regulatory

**eLife digest** Inside the nucleus of a cell, the DNA is wound around histone proteins. This forms a structure called chromatin that allows the long DNA strands to fit inside the cell. Variations in chromatin structure also help the cell to control the functional properties of DNA. For example, a large proportion of chromatin in the cell is in the form of heterochromatin, which is very densely packed, and is associated with many roles such as gene silencing and keeping DNA intact during reproduction.

Many animals and plants have two copies of each DNA molecule: one inherited from the mother, and one from the father of the organism. Reproductive cells undergo a process called recombination when they form, where the matching copies of each DNA molecule break in a number of places and rejoin to form a new 'blend' of their mother's and their father's DNA, which is passed on to their own offspring. In contrast, most heterochromatin is inherited without recombining, preserving it in an unaltered form. This is important since recombination in heterochromatin can create genetic abnormalities.

Adding small chemical modifications—such as methyl groups—to the histone proteins at the core of the chromatin can change how the DNA is packed. However, the histone modifications that yield different chromatin structures, and the effect of these modifications, are not very well understood.

Papazyan et al. have taken advantage of a distinct feature of the protozoan *Tetrahymena thermophila*: a single-celled organism that divides its chromatin into two different nuclei. The smaller micronuclei contain only heterochromatin, and Papazyan et al. discovered that the histone H3 protein in the micronuclei is modified by methyl groups at a specific site that had not been studied before. Furthermore, this protozoan makes more of these modifications when it reproduces. An enzyme called Ezl3p adds these methyl groups, and without this enzyme *T. thermophila* reproduces more slowly and has offspring that are less likely to survive and more likely to be infertile. Papazyan et al. provide evidence that these characteristics arise because the cells without the histone modification are unable to prevent DNA breaks from occurring in heterochromatin during recombination.

The same histone modification also occurs when the microscopic worm *Caenorhabditis elegans* reproduces, suggesting that this method of DNA protection has been conserved throughout evolution. Papazyan et al. propose that the histone modification may prevent another enzyme that induces DNA breaks from accessing the heterochromatin in reproductive cells; but more work is required to support this hypothesis.

These findings reveal the importance of a new histone modification during reproduction, and could provide new directions for infertility research.

roles in a wide variety of biological processes including DNA replication, recombination, and repair (*Dernburg et al., 1996*; *Lukas et al., 2011*; *Alabert and Groth, 2012*). Accordingly, characterizing factors that regulate heterochromatin formation and maintenance is important for advancing studies on health and human disease, as well as for appreciating basic biology.

While heterochromatin was described cytologically in 1928 as intensely stained parts of chromosomes that remain condensed over the cell cycle (*Heitz, 1928*), molecular definitions of heterochromatin remain elusive. Of particular interest is the complement of posttranslational modifications (PTMs) that occur on histone proteins within this transcriptionally 'off' environment. Studies of histone PTMs such as methylation, acetylation, or phosphorylation have shown they assist in regulation of chromatin activity, which has helped usher in a modern understanding of different varieties or subdomains of this compact chromatin state (*Strahl and Allis, 2000*; *Turner, 2000*). 'Constitutive' heterochromatin is found at structural or highly repetitive stretches of the genome such as pericentric or subtelomeric regions, is enriched in Su(var) (suppressors of position effect variegation) proteins and trimethylation on lysine 9 of histone H3 (H3K9me3) (*James et al., 1989*; *Bannister et al., 2001*; *Jacobs et al., 2001*; *Peters et al., 2001*). Regions of 'facultative' heterochromatin are also condensed; however, they are enriched in H3K27me3, and can become activated or silenced in response to different cellular environments, such as the inactive X chromosome in mammalian females (*Simon and*

*Kingston, 2009*). Despite being generally repressed, much of heterochromatin can be transcribed and processed into non-coding RNA, which in turn has been linked to trans-generational deposition of histone modifications and epigenetic silencing (*Djebali et al., 2012*; *Gu et al., 2012*). Further characterization of PTMs associated with heterochromatin will be necessary to appreciate its subtle distinguishing features and overall regulation.

To gain insight into histone PTM states enriched in heterochromatin, we took advantage of the nuclear dimorphism characteristic of the ciliated protozoan *Tetrahymena thermophila*. While most cells from higher eukaryotes have a single nucleus, where euchromatin and heterochromatin are intermingled on contiguous stretches of chromosomes, *Tetrahymena* maintain two functionally distinct nuclei within a common cytoplasm: a transcriptionally inert, heterochromatic micronucleus and a transcriptionally active, euchromatic macronucleus (*Figure 1A*). The micronucleus contains the complete *Tetrahymena* genome whereas the genomic complexity of the macronucleus (derived from the micronucleus during sexual reproduction) is reduced by ~33% through programmed DNA elimination (http://www.broadinstitute.org/annotation/genome/Tetrahymena/MultiHome.html). Most of the eliminated, micronuclear-limited sequences are repetitive, centromeric, or otherwise non-coding DNA (*Chalker, 2008*; *Schoeberl et al., 2012*).

In this study, we extracted histone H3 from highly purified *Tetrahymena* micronuclei and used ETD (electron transfer dissociation) mass spectrometry to screen for combinatorial histone PTMs on single H3 peptides (*Mikesh et al., 2006*; *Taverna et al., 2007b*; *Young et al., 2009*). We identified H3K23me3 as a micronucleus-specific PTM that exclusively co-occurs with H3K27 methylation on the same H3 molecule. Using an antibody specific for H3K23me3, we found this mark becomes highly enriched during early stages of meiosis in *Tetrahymena*. Moreover, the loss of H3K23me3 resulted in developmental lags of meiosis, delayed transcription of germline-specific non-coding RNAs, mislocalization of meiotic DNA double strand breaks (DSBs), and reduction of progeny viability. Finally, we found that H3K23me3 levels also increase in *C. elegans* germ cells upon meiotic entry. Together, our data suggest H3K23me3 is a conserved heterochromatic histone PTM strongly associated with meiosis, and misregulation of this modification may be linked to problems with germline recombination and reproductive fitness.

## Results

### Purification of histone H3 from *Tetrahymena* heterochromatin

*Tetrahymena* maintain two functionally distinct nuclei that also differ in histone PTM states (*Vavra et al., 1982*). Our previous studies have shown that the germline micronucleus is relatively enriched with H3K27me3 and completely lacking in the euchromatic PTM H3K4me3 (*Liu et al., 2007*; *Taverna et al., 2007b*). In contrast, the somatic macronucleus is positive for both the H3K27me3 and H3K4me3, although these PTMs localize to distinct sub-nuclear domains (*Figure 1A*). We obtained highly purified H3 from the heterochromatic germline by efficiently isolating micronuclei away from macronuclei (*Figure 1B*), extracting total micronuclear histones, and resolving them by reverse-phase HPLC (*Figure 1C*). As expected, micronuclear H3 occurs in two electrophoretically distinct forms in vivo (*Figure 1C*, inset): the slow form of micronuclear H3 (H3$^S$) is the same sequence as macronuclear H3; and the fast form (H3$^F$) is derived from H3$^S$ during a micronucleus-specific proteolytic processing step that removes the first six amino acids from the H3$^S$ N-terminus (*Allis et al., 1980*).

### Combinatorial modification states of single histone H3 tails purified from heterochromatic micronuclei

To determine histone PTM states on micronuclear histone H3, H3$^S$ and H3$^F$ were analyzed using pH gradient hydrophilic interaction liquid chromatography (HILIC) directly coupled to high-resolution mass spectrometry using electron transfer dissociation (ETD; subsequently referred to as HILIC-MS/MS) (*Young et al., 2009*). HILIC improves the separation and interrogation of large, hydrophilic, and highly modified H3 peptides. Additionally, ETD helps identify novel combinations of histone PTMs that simultaneously occur on highly-charged, long (>20 amino acids) peptides by providing complete to near complete fragment ion coverage (*Coon et al., 2005*; *Thomas et al., 2006*; *Garcia et al., 2007b*; *Taverna et al., 2007b*; *Young et al., 2009*).

Our HILIC-MS/MS analysis revealed that, similar to condensed chromatin in higher eukaryotes (*Jeppesen and Turner, 1993*), micronuclear H3$^S$ and H3$^F$ were collectively hypoacetylated, deficient in

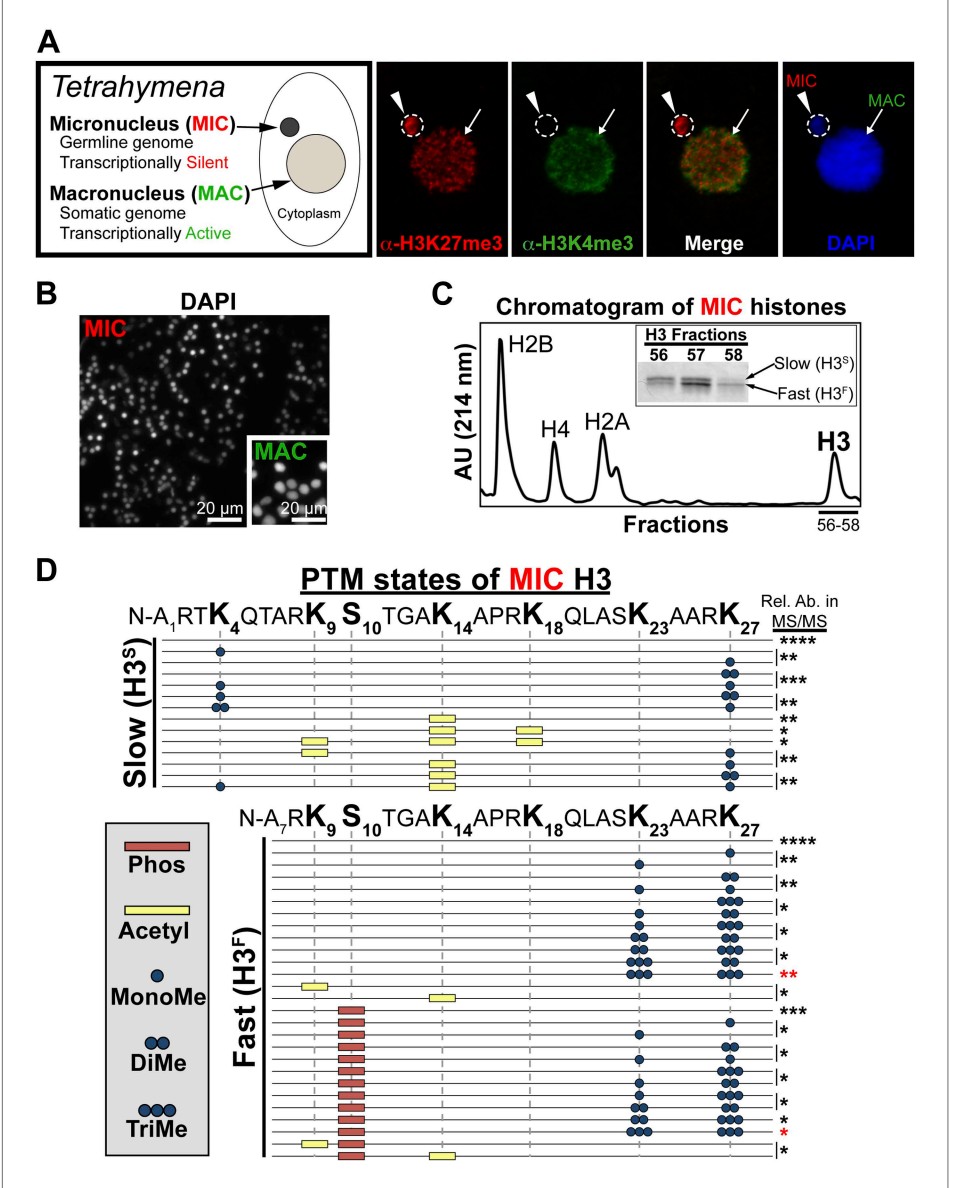

**Figure 1**. Purification of histone H3 from heterochromatic micronuclei and analysis of the associated PTM states. (**A**) Co-immunofluorescence staining of Tetrahymena. (**B**) Following cell lysis, micronuclei were efficiently separated from macronuclei by differential centrifugation. Shown is a representative micronuclear fraction that was used in subsequent purification steps. (**C**) Chromatogram of micronuclear core histones resolved by RP-HPLC. Inset shows fractions containing two micronuclear histone H3 species, H3$^S$ ('S' for electrophoretically slow) and H3$^F$ ('F' for electrophoretically fast), which are stained with Coomassie following SDS-PAGE. (**D**) Summary of modified forms of micronuclear histone H3 N-terminal peptides (residues 1–50 for H3$^S$ and 7–50 for H3$^F$) from vegetatively growing Tetrahymena as detected by HILIC-MS/MS. Sequences of H3$^S$ and H3$^F$ are shown up to K27 since no modifications were detected on residues 28–50. Each row in the table represents a single, discrete, and uniquely modified H3 N-terminal peptide, and each dotted horizontal line beneath a bolded residue represents a site where the specified modification was identified. The relative abundance of each peptide, or multiple isobaric peptides identified within the same spectrum (grouped by brackets on the right), is highlighted by the number of asterisks. For example, peptides with four asterisks are more abundant than peptides with one asterisk. Forms of the H3$^S$(1–50) peptide with four, five, and six methyls were detected, but due to the isobaric character of each group of peptides and the low levels of these species it was not possible to determine the locations of the methyl modifications. See also *Figure 1—figure supplement 1* for supporting antibody-based validation of the presence of H3K23

*Figure 1. Continued on next page*

*Figure 1. Continued*

mono- and dimethylation in Tetrahymena, and *Figure 1—source data 1* for supporting mass spectrometry data. Abbreviations: MonoMe, monomethylation; DiMe, dimethylation; TriMe, trimethylation.

The following source data and figure supplement is available for figure 1:

**Source data 1**. Shown are the tandem mass spectrometry data for the N-terminus of histone H3 in the unmodified, and dually modified H3K23me3/H3K27me3 states.

**Figure supplement 1**. Presence of H3K23me1 and H3K23me2 in Tetrahymena.

H3K4me, and enriched for H3K27me (*Figure 1D*). The multiply acetylated, monomethylated H3$^S$K4, and monomethylated H3$^S$K27 species likely represent newly deposited histones in micronuclei (*Sobel et al., 1995*; *Nightingale et al., 2007*; *Gao et al., 2013*). As expected from our previous studies, H3K9me was not detected on any species of vegetative micronuclear H3, while H3K27me3 and the mitotic PTM, histone H3 serine 10 phosphorylation (H3S10ph), were only detected on H3$^F$ (*Wei et al., 1998*; *Taverna et al., 2007b*). Unexpectedly, we detected methylation on H3K23, which was highly enriched on H3$^F$ and mostly occurred in combination with H3K27me (*Figure 1D*). To validate the presence of H3K23 methylation within *Tetrahymena*, we used antibodies reported to recognize H3K23me1 and H3K23me2, to show that these PTMs are detectable on micronuclear H3 (*Figure 1—figure supplement 1A,B*).

While only a few studies have identified methylation on H3K23 through Edman degradation and mass spectrometry in somatic cells, the biological relevance of H3K23me is not clear (*Waterborg, 1990*; *Garcia et al., 2007a*; *Liu et al., 2010*). We focused on H3K23me3 since it was highly associated with H3K27me3 and the H3K23me3/H3K27me3 PTM state was most abundant out of all the dually methylated H3K23/H3K27 states (*Figure 1D*, red asterisks). To investigate the function of this undercharacterized histone PTM, we next developed antibodies specific for H3K23me3 (α-H3K23me3), as none were available at that time.

## Generation of a polyclonal antibody specific for H3K23me3

Antibodies against H3K23me3 were generated using a synthetic H3 peptide containing residues 18–36 and the H3K23me3 modification (*Figure 2A*, underlined). ELISA analysis of the affinity-purified antibody was then carried out to test for specificity. As shown in *Figure 2B*, the antibody was highly specific for peptides containing the H3K23me3 mark. Furthermore, preincubation of the antibody with excess H3K23me3 peptide, but not H3K4me3, H3K9me3, or H3K27me3 peptides, competed away immunoreactivity for H3K23me3 (*Figure 2C*). The antibody was also tested against distinctly methylated forms of H3K23 and only the H3K23me2 peptide was weakly recognized (*Figure 2—figure supplement 1*). Notably, α-H3K23me3 antibodies did not bind to the H3K9me3 peptide, even though the neighboring sequence of H3K9 is very similar to that of H3K23 (*Figure 2A*, yellow boxes). Thus, the antibody is highly specific for H3K23me3, and importantly, the presence of H3K27me3 did not occlude it from binding to the dual H3K23me3/H3K27me3 PTM state we identified in *Tetrahymena*.

## H3K23 trimethylation is specific to the germline micronucleus in *Tetrahymena*

Our HILIC-MS/MS analysis revealed that H3K23me3 is specifically enriched on H3$^F$ of micronuclei. To confirm this, we probed histones from highly purified micronuclei and macronuclei using α-H3K23me3. The results showed that H3K23me3 was only present on the proteolytically processed H3$^F$ from micronuclei, but not H3$^S$ from micronuclei or macronuclei (*Figure 2D*). The antibody also failed to detect recombinant H3 (rH3) indicating that the methylated state of H3K23 is required for effective epitope recognition. Indirect immunofluorescence using α-H3K23me3 further confirmed that H3K23me3 was exclusively localized to germline micronuclei in *Tetrahymena* (*Figure 2E*). Coupled with the fact that H3K23me3 is not detectable on macronuclear H3 (*Taverna et al., 2007b*), our results indicate that this PTM is exclusive to the *Tetrahymena* germline.

## H3K23me3 is greatly increased in micronuclei during meiosis

Since H3K23me3 is restricted to the germline micronucleus during vegetative growth, we wanted to explore the possibility that H3K23me3 plays an important role during conjugation, when micronuclear

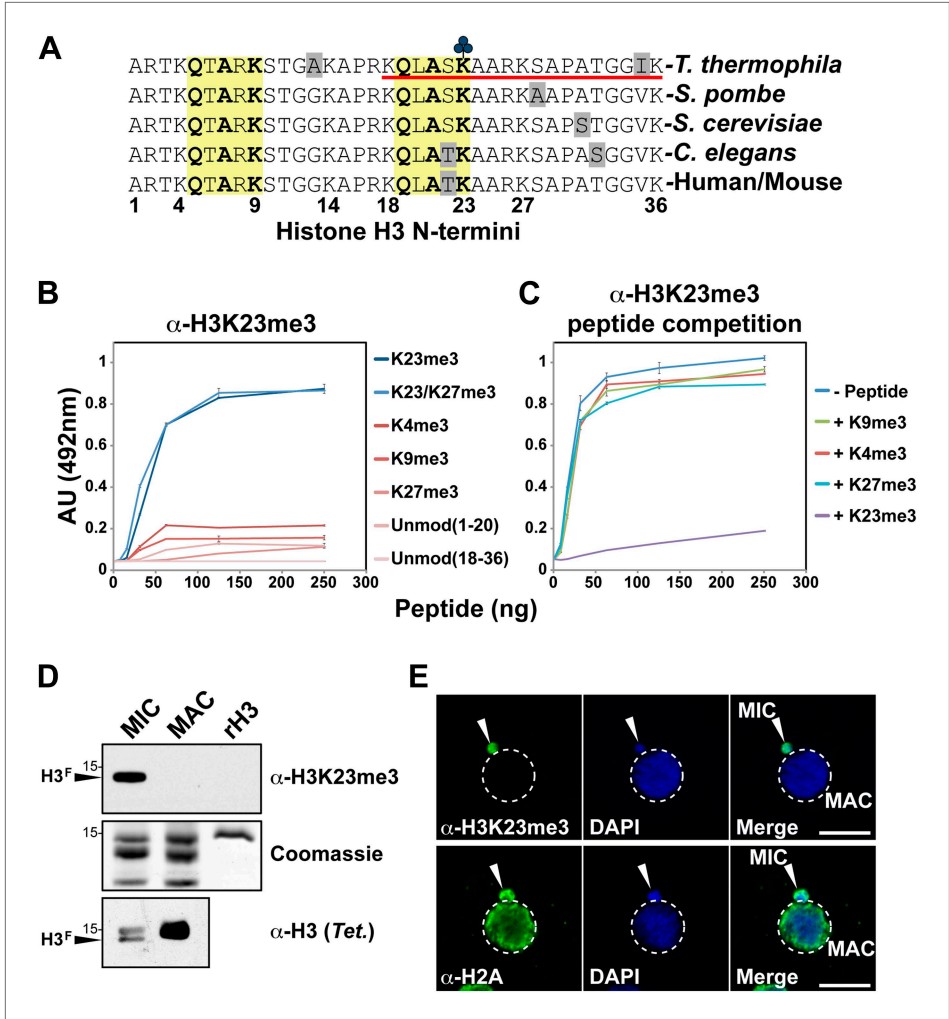

**Figure 2**. Development of an H3K23me3-specific antibody. (**A**) Alignment of the first 36 residues of histone H3 from the indicated organisms. An H3 peptide spanning amino acids 18–36 (red underline) and trimethylated on K23 (blue clover) were synthesized and used as the immunogen for antibody generation in rabbits. Yellow boxes highlight the conserved QXAXK motif associated with H3K9 and H3K23. (**B**) ELISA analysis of α-H3K23me3 binding to the indicated peptides. See also *Figure 2—figure supplement 1*. (**C**) ELISA analysis of α-H3K23me3 binding to H3K23me3(18–36) peptide in the presence of excess competing peptides. (**B–C**) Data points represent mean ± SD of triplicate samples from a representative experiment. (**D**) Western blot analysis of α-H3K23me3 specificity using micronuclear and macronuclear extracts, and recombinant human H3 (rH3). (**E**) Indirect immunofluorescence staining of vegetatively growing Tetrahymena using either α-H3K23me3 or α-H2A (control). White arrowheads point to micronuclei. Scale bar, 10 μm.

The following figure supplement is available for figure 2:

**Figure supplement 1**. Affinity of α-H3K23me3 for distinct methylation states of H3K23.

function is essential. During conjugation, parental micronuclei give rise to both macronuclei and micronuclei in progeny, while the parental macronuclei degrade. To accomplish this, germline micronuclei undergo meiotic and mitotic divisions in a series of well-defined developmental stages (*Martindale et al., 1982*; *Chalker, 2008*). Importantly, conjugation proceeds in a synchronous manner, allowing the study of histone PTMs at specific developmental windows.

Whole-cell-extracts collected at key time points during conjugation revealed that H3K23me3 levels greatly increase on H3[F], peaking by hour ~4, and diminishing to levels observed in vegetative cells by hour 8 (*Figure 3A*). Interestingly, up-regulation of H3K23me3 early in conjugation implied a

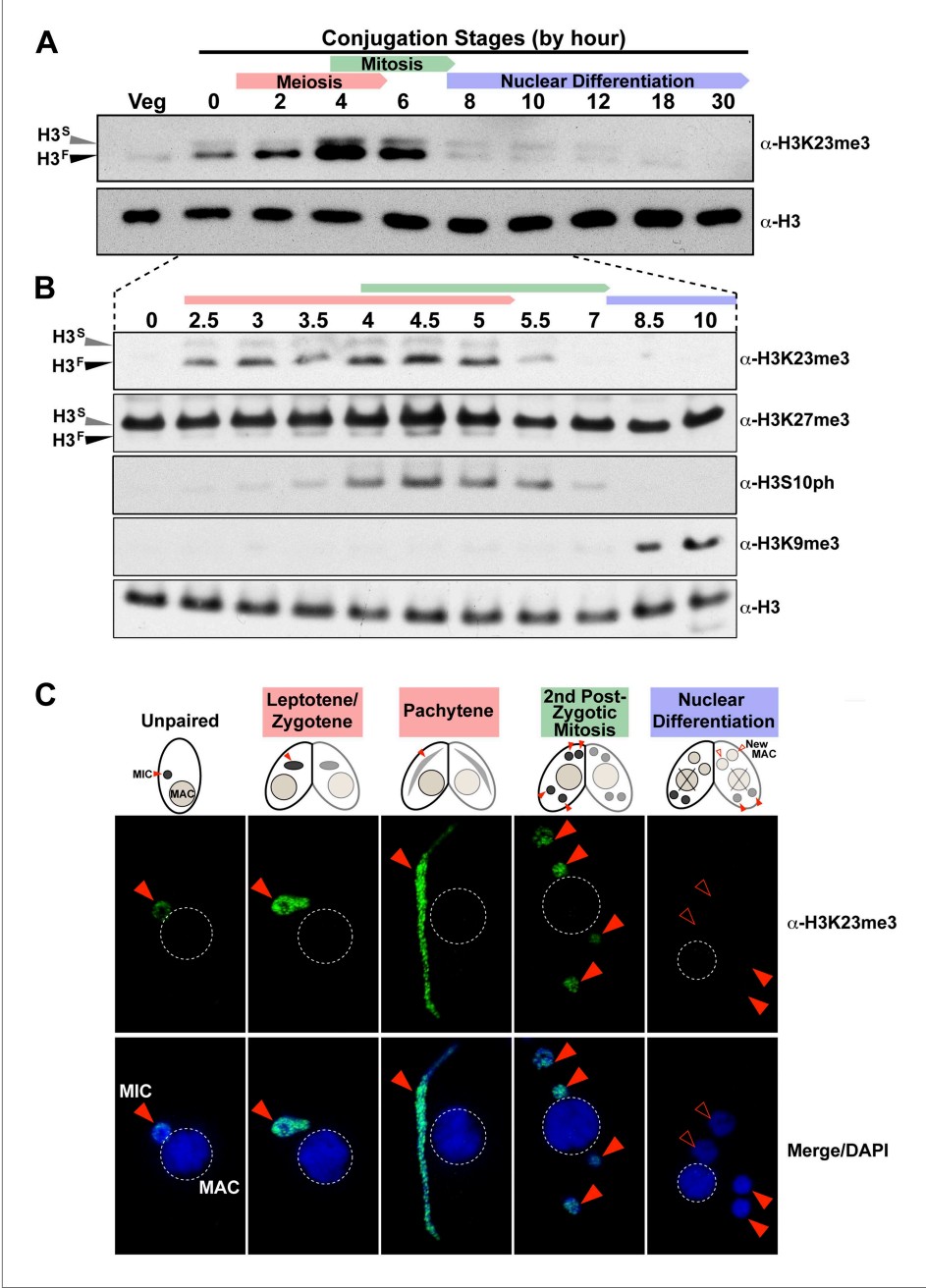

**Figure 3**. H3K23me3 levels are dramatically increased during early meiosis. (**A** and **B**) Whole-cell-extracts of wild type cells were collected at the indicated time points during conjugation, resolved on SDS-PAGE gels, and analyzed by Western blotting. Color-coded arrows highlight the period of the specified developmental stage. See also *Figure 3—figure supplement 1A*. (**C**) Indirect immunofluorescence analysis of conjugating Tetrahymena using α-H3K23me3. Red arrowheads point to micronuclei. During nuclear differentiation, the parental macronucleus (labeled with 'X' in the cartoon) degrades while two post-zygotic nuclei begin to differentiate to new macronuclei (open red arrowheads) and the other two post-zygotic nuclei remain as micronuclei. See also *Figure 3—figure supplement 1B*.

The following figure supplement is available for figure 3:

**Figure supplement 1**. Combinatorial modification states of micronuclear H3 from conjugating Tetrahymena.

connection to meiosis, which precedes mitosis. Indeed, H3K23me3 levels increase before the mitosis-associated PTM H3S10ph becomes enriched (*Figure 3B*). While increased H3K23me3 coincided with meiotic entry, the PTM was still present, although much reduced, during stages of mitosis as evidenced by our Western blot (*Figure 3B*), immunofluorescence (*Figure 3C*), and HILIC-MS/MS data (*Figure 3—figure supplement 1A*). A small amount of H3K23me3 was detected on H3$^S$ during conjugation (*Figure 3A*), but our immunofluorescence data (*Figure 3C*) suggest this is restricted to the micronucleus and at levels below the detection limit of HILIC-MS/MS (*Figure 3—figure supplement 1A*).

The physical link between H3K23me3 and H3K27me3 prompted us to examine these modifications on H3$^F$ during meiosis. While H3$^S$K27me3 levels in the macronucleus remain constant throughout conjugation (*Liu et al., 2007*), we found that H3$^F$K27me3 levels in the micronucleus increased and diminished concurrently with H3K23me3 levels (*Figure 3B*). Our HILIC-MS/MS analysis of micronuclear histone H3 purified from conjugating cells also showed that up-regulated H3K23me3 remained physically linked to H3K27me3 (*Figure 3—figure supplement 1A*). In contrast to the coordinate increase of H3K23me3 and H3K27me3 observed during meiosis, H3K9me3, which specifically emerges during nuclear differentiation (*Liu et al., 2007*; *Taverna et al., 2002*), was only detected after levels of H3K23me3 diminished (*Figure 3B*). The fact that low levels of H3K23me3 detected during nuclear differentiation are present in micronuclei and not the H3K9me3-enriched developing macronuclei (*Figure 3—figure supplement 1B*), further defines distinct functions within heterochromatin for these PTMs. Taken together, these findings suggest that H3K23me3 may play an important new role during meiosis and heterochromatin formation.

## An Enhancer of zeste homolog in *Tetrahymena* is required for K23 methylation in the germline micronucleus

To provide biological insight about H3K23me3, we sought to identify genetic knockouts that would disrupt this PTM. As no H3K23me3 methyltransferase was known, we focused on investigating *Tetrahymena* homologs of the H3K27me3 'writer' Enhancer of zeste reasoning that the loss of H3K27me3 could also disrupt H3K23me3 levels on the same H3 protein. We previously identified three Enhancer of zeste-like (Ezl) proteins in *Tetrahymena*: Ezl1p, Ezl2p, and Ezl3p (*Liu et al., 2007*). While Ezl1p and Ezl2p are required for H3K27me3 during distinct life cycles, the protein target of Ezl3p is unknown (*Figure 4—figure supplement 1A*) (*Liu et al., 2007*; *Chung and Yao, 2012*).

Whole-cell-extracts from vegetatively growing *EZL* knockouts (Δ*EZL1*, Δ*EZL2*, or Δ*EZL3*) were screened for changes in H3K23me3 by Western blotting. While H3K23me3 was present at wild type levels in Δ*EZL1* cells, their levels were significantly reduced in Δ*EZL2* cells and completely undetectable in Δ*EZL3* cells (*Figure 4A–B*). In addition to H3K23me3, Ezl3p expression was required for H3K23me1 and H3K23me2 (*Figure 4—figure supplement 1B,C*). In line with these results, mass spectrometric analysis of all micronuclear core histones from Δ*EZL3* cells revealed that only H3K23 methylation is lost in these mutants (*Figure 4—figure supplement 1,F*). Together, these results implicate H3K23 as the likely target for Ezl3p activity in micronuclei. Ezl3 expression was strongly upregulated during early conjugation, further supporting an integral role for Ezl3 in meiosis and H3K23 methylation (*Figure 4—figure supplement 1D*).

To strengthen the biochemical link between Ezl3p and H3K23 methylation, we reintroduced catalytic Ezl3p activity in Δ*EZL3* cells. As a control, we included a catalytically inactive mutant, Ezl3p$^{H599A}$, harboring a point mutation in the asparagine-histidine-serine catalytic triad of the Ezl3p SET domain. Mutation of any residue in this highly conserved triad has been shown to abolish enzymatic activity of SET domain-containing histone methyltransferases (*Jacobs et al., 2002*). As shown in *Figure 4C*, expression of wild type Ezl3p, but not Ezl3p$^{H599A}$, rescued the levels of H3K23me3 in Δ*EZL3* cells. While these results demonstrate catalytically active Ezl3p is necessary and sufficient for the deposition of H3K23me3 in *Tetrahymena* in vivo, we were not able to purify Ezl3 in amounts sufficient to examine Ezl3 methylation of H3K23 in vitro.

## A developmental lag is observed in conjugating *Tetrahymena* that lack Ezl3p and H3K23 methylation

We proceeded to analyze Δ*EZL3* cells for phenotypes to better understand how the loss of H3K23me3 impacts *Tetrahymena*. In log phase, the growth rate of Δ*EZL3* cells was about 20% slower than wild type cells (*Figure 5—figure supplement 1*). However, since H3K23me3 becomes dramatically upregulated during early meiosis, we proceeded to analyze Δ*EZL3* cells for meiotic phenotypes.

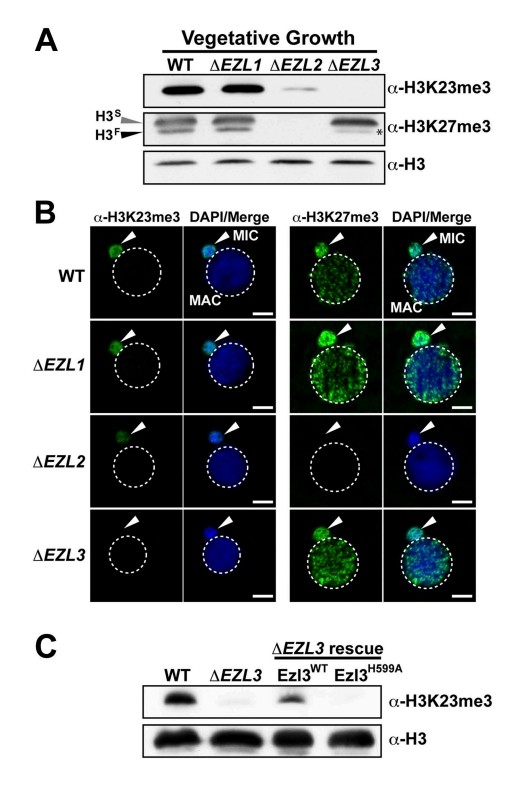

**Figure 4**. An Enhancer of zeste homolog is required for H3K23 methylation in Tetrahymena. (**A**) Western blot analysis of whole-cell-extracts and (**B**) indirect immuno-fluorescence staining of wild type (WT), *ΔEZL1*, *ΔEZL2*, and *ΔEZL3* cells grown under vegetative conditions. The asterisk highlights the reduction of H3^FK27me3 in *ΔEZL3* cells. White arrowheads point to micronuclei. Scale bar, 5 μm. (**C**) Ezl3p expression was reconstituted in *ΔEZL3* cells. As a control, a catalytically inactive mutant generated by a His (599) to Ala mutation was included. See also *Figure 4—figure supplement 1*.

The following figure supplement is available for figure 4:

**Figure supplement 1**. Characterization of *EZL3* in vegetatively growing and conjugating Tetrahymena.

Whole-cell-extracts of conjugating wild type or *ΔEZL3* cells were collected at indicated time points and analyzed by Western blotting. H3S10ph (mitotic marker) and Pdd1p (*Taverna et al., 2002*) levels were used as markers for progression of early and late conjugation, respectively (*Figure 5A*). Remarkably, a significant lag was detected during early conjugation in *ΔEZL3* cells, exemplified by a delay in the onset of H3S10ph. Moreover, the developmental delay of *ΔEZL3* cells became more pronounced during nuclear differentiation, as high-lighted by levels of Pdd1p at later stages of conjugation (*Figures 5A*, 18hr–48hr). Cytologic comparison of wild type and *ΔEZL3* mating pairs showed that the initial lag during conjugation began at the leptotene/zygotene stages of meiotic prophase (*Figure 5B*). *ΔEZL3* cells also showed delayed formation of piRNA-like scan-RNA (*Figure 5C*), a class of non-coding RNAs intricately involved with heterochromatin forma-tion and DNA elimination at the nuclear differen-tiation stage of conjugation (*Mochizuki et al., 2002*; *Liu et al., 2007*). Despite a pronounced lag in the completion of meiosis, nuclear differentia-tion, and scanRNA formation, *ΔEZL3* cells even-tually proceeded to terminal stages of conjugation (*Figure 5B*).

## Loss of Ezl3p and H3K23 methylation in *Tetrahymena* is associated with ectopic DNA double-strand breaks

The significant meiotic lag in conjugating *ΔEZL3* cells suggested meiosis-induced DNA damage or repair pathways were affected in these mutants. During meiosis, the programmed formation of DNA double-strand breaks (DSB) facilitates loci pairing and homolog segregation and promotes recombination and genetic diversity (*Lichten and de Massy, 2011*). Misregulation of DSB pathways within the germline can reduce genomic stability, the production of viable gametes, and progeny viability (*Brachet et al., 2012*). Conveniently, phos-phorylation on the C-terminus of histone H2A.X (γH2A.X) is a well-known indicator of the presence of DSBs and the ensuing DNA damage response, in a myriad of organisms including *Tetrahymena*, and α-γH2A.X is also used to assess meiotic pro-gression (*Song et al., 2007* and references therein). Therefore, we used α–γH2A.X to investigate defects in the localization of DSBs in meiotic *ΔEZL3 Tetrahymena* cells. Micronuclear chromosomes in early stages of meiosis become highly organized—centromeres and highly-repetitive pericentric regions (DAPI-poor) segregate to one pole of the elongating micronucleus, while telomeres and genic regions (DAPI-rich) localize to the opposite pole (*Figure 6A*) (*Loidl and Scherthan, 2004*; *Mochizuki et al., 2008*).

As illustrated in *Figure 6A*, DSBs in wild type micronuclei initiated early in meiosis (leptotene/zygotene) within the DAPI-rich genic region, becoming distributed throughout the micronucleus by pachytene. In striking contrast, while some DSB signal in early meiotic *ΔEZL3* micronuclei could be detected in the DAPI-rich genic region, the vast majority of DSB signal covered the DAPI-poor,

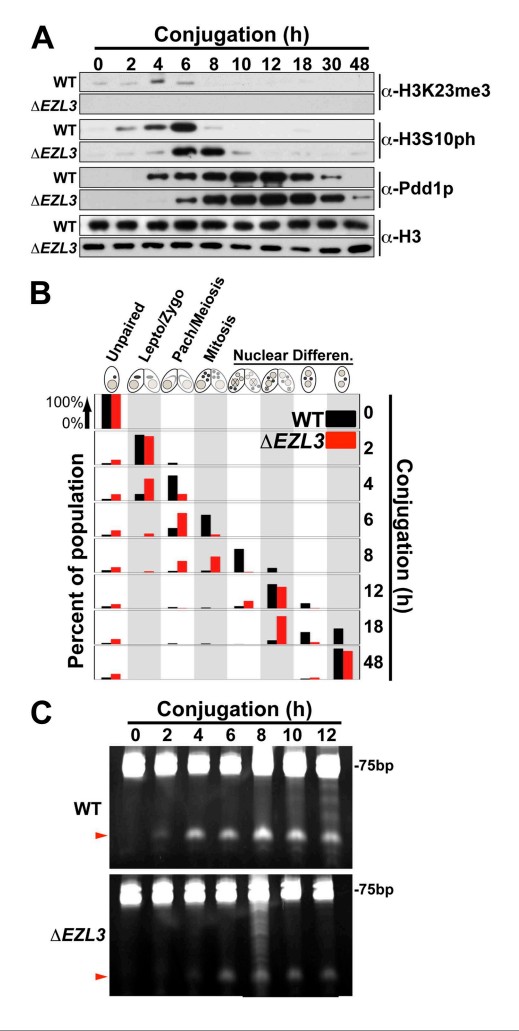

**Figure 5**. Meiotic progression is disrupted in Tetrahymena lacking Ezl3p and H3K23 methylation. (**A**) Western blot analysis of whole-cell-extracts of wild type and *ΔEZL3* cells progressing through conjugation. H3S10ph and Pdd1p were used as markers of early and late conjugation, respectively. (**B**) Conjugating wild type (black) and *ΔEZL3* (red) cells was fixed at the indicated time points and analyzed by microscopy. Cytological hallmarks associated with each developmental stage (cartoons on top) were used to assess their progression through conjugation. This analysis revealed a developmental lag in early meiotic *ΔEZL3* cells (4 hr). (**C**) Total RNA, extracted from wild type and *ΔEZL3* cells during conjugation, was resolved on denaturing acrylamide gels and stained with ethidium bromide. Red arrowheads point to the location of piRNA-like scanRNA. See also *Figure 5—figure supplement 1* for additional *ΔEZL3* phenotype information.

The following figure supplement is available for figure 5:

**Figure supplement 1**. Vegetatively growing *ΔEZL3* cells have a slow growth phenotype.

centromeric pole of micronuclei (*Figure 6A*). In meiotic *ΔEZL3* cells, the increased DSB signal over the DAPI-poor, centromeric pole of micronuclei remained for the majority of the meiotic lag, although when mutant cells finally entered pachytene, γH2A.X localization resembled wild type distribution (*Figure 6A*). Interestingly, DSBs were not mislocalized in *ΔEZL2* cells (lacking H3K27me3), suggesting that only cells completely lacking H3K23me3 have mislocalized meiotic DSBs (*Figure 6B*).

Centromeric H3 (α-Cna1p) was similarly positioned in both wild type and *ΔEZL3* cells, showing that the shift in DSB positioning in *ΔEZL3* cells was not the result of massive chromatin reorganization within the nuclei (*Figure 6C*) (*Cervantes et al., 2006*). Further evidence in support of this interpretation was provided by immunoFISH assays combining α-γH2A.X and a DNA probe against the Tlr1 element (*Wells et al., 1994*; *Reddy et al., 2008*). A number of micronuclear-limited sequences, including Tlr1, are repetitive in nature and are enriched in centromere-proximal regions, whereas genes are enriched more distally on the chromosome arms (EP Hamilton and RS Coyne, personal communication). The Tlr1 probe we generated was highly specific for micronuclei (*Figure 6—figure supplement 1*). In wild type meiotic micronuclei, we detected minimal overlap between the α-γH2A.X and Tlr1 probe signals. However in *ΔEZL3* cells, these signals were highly overlapping and enriched near the centromeric pole of meiotic micronuclei, further suggesting that DSBs in cells lacking H3K23me3 are mislocalized to pericentric regions (*Figure 6D*).

Misregulation of DSB pathways within the germline reduces genomic stability, the production of viable gametes, and progeny viability (*Brachet et al., 2012*). Therefore, we tested whether the loss of H3K23me3 and ectopic DSB formation translated to reduced progeny viability in *Tetrahymena*. Although parental *ΔEZL3* cells reach terminal cytological stages of conjugation (*Figure 5B*, 48 hr), the progeny viability was significantly impaired (*Figure 6E* and *Figure 6—source data 1*). Furthermore, fertility was also negatively affected when surviving *ΔEZL3* progeny were backcrossed to wild type cells (data not shown).

Taken together, we hypothesize that H3K23me3 helps mediate the formation of a higher-order chromatin structure during meiosis that is distinct from H3K9me3-associated heterochromatin. Our observations further suggest that H3K23me3-enriched heterochromatin helps to protect pericentric, repetitive genetic sequences from meiotic

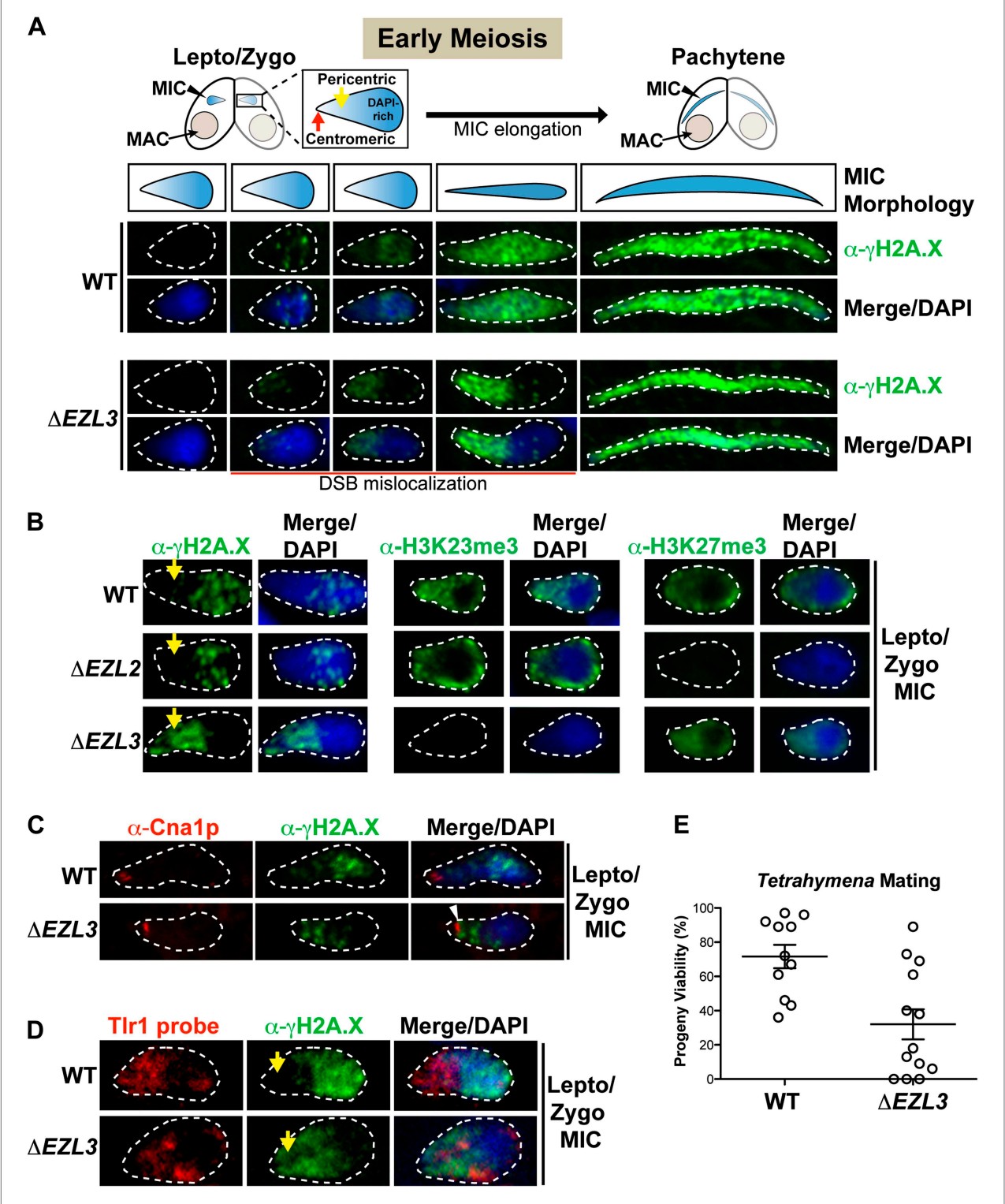

**Figure 6**. Loss of K3K23me3, but not H3K27me3, during meiosis leads to increased DSBs in pericentric heterochromatin. (**A**) Cartoon depiction of the morphological changes to micronuclei during early meiotic prophase. At leptotene/zygotene, chromatin is spatially organized within the micronucleus with a DAPI-rich pole, a DAPI-poor pericentric region (yellow arrow), and a centromeric pole (red arrow). Below the illustrations are images of indirect immunofluorescence stains of leptotene/zygotene micronuclei using α-γH2A.X. Representative images are shown from at least three independent experiments. The micronuclei presented in this figure represent stages II–III/IV of meiotic prophase as reported previously (**Sugai and Hiwatashi, 1974**).
*Figure 6. Continued on next page*

*Figure 6. Continued*

(**B**) Indirect immunofluorescence staining of leptotene/zygotene micronuclei from wild type, *ΔEZL2,* and *ΔEZL3* cells using α-γH2A.X, α-H3K23me3, and α-H3K27me3. Yellow arrows point to the pericentric region of meiotic micronuclei. (**C**) Images of leptotene/zygotene micronuclei from wild type and *ΔEZL3* cells co-stained with α-Cna1p (centromeric H3) and α-γH2A.X. Note the white arrowhead at the tight, but non-overlapping junction between DSBs and the centromere in *ΔEZL3* micronuclei. (**D**) ImmunoFISH staining using α-γH2A.X and a DNA probe against the pericentric marker Tlr1. See also *Figure 6—figure supplement 1*. (**E**) Percentage of genetic progeny resulting from mating of wild type lines vs isogenic *ΔEZL3* lines. Shown in the graphs are mean ± SEM (p < 0.005 as calculated by a two-tailed *t*-test). See also *Figure 6—source data 1* for the wild type and mutant crossing data, and *Figure 6—figure supplement 2* for a model proposing how H3K23me3 may function during meiosis.

The following source data and figure supplements are available for figure 6:

**Source data 1**. Progeny viability is significantly reduced in *ΔEZL3* cells.

**Figure supplement 1**. The Tlr1 probe is specific for germline chromatin.

**Figure supplement 2**. A model proposing that H3K23me3 mediates the formation of a higher-ordered heterochromatin structure to limit DSBs at pericentric sequences during early meiosis.

DSBs, which are catalyzed by the opportunistic and highly conserved Spo11 nuclease (*Figure 6—figure supplement 2*).

## H3K23me3 in nematodes is largely confined to meiosis, and is distinct from H3K9me3 and H3K27me3

Because the H3K23 epitope is highly conserved across eukaryotes (*Figure 2A*), we next tested whether the H3K23me3 PTM was detectable in *C. elegans*. This model organism has provided important cytological and genetic insights into gametogenesis/germ cell differentiation, as well as processes regulating mitotic and meiotic checkpoints (*Kimble and Crittenden, 2007*). We initially examined nuclear extracts from whole *C. elegans* embryos, because their nuclei can be easily purified, making possible a direct comparison of H3K23me3 levels between *C. elegans* and *Tetrahymena*. Comparative Western blot analysis of whole-embryo nuclear extracts showed that H3K23me3 was present in *C. elegans* chromatin, although at a much lower abundance than in *Tetrahymena* micronuclear chromatin (*Figure 7A*).

Given that H3K23me3 is present in *C. elegans*, we next wanted to determine if this mark is upregulated during meiosis. The *C. elegans* gonad is essentially organized into two physically and functionally distinct regions. Germ cells near the distal tip cell of the gonad are maintained as a mitotically dividing population through Notch-like (*glp-1*) signaling; however, as germ cells migrate proximally, they enter meiosis and differentiate into sperm or oocytes (*Figure 7B*) (*Colaiacovo et al., 2003*; *Kimble and Crittenden, 2007*). By indirect immunofluorescence, H3K23me3 was nearly absent in the mitotic region of the gonad, although a small number of mitotic nuclei in this region were positive for the PTM (*Figure 7C*). In striking contrast, H3K23me3 staining became robust in meiotic cells within the gonad, beginning at the leptotene/zygotene transition zone that marks meiotic entry, and plateauing through the rest of meiotic prophase (*Figure 7C* and *Figure 7—figure supplement 1A*). Compared to H3K27me3, H3K9me3, and H3K4me3 distribution in the gonad, H3K23me3 was the only PTM that became enriched as germ cells entered the transition zone (*Figure 7D* and *Figure 7—figure supplement 1B*). Under high magnification, H3K23me3 and other PTMs also exhibited comparatively different sub-nuclear staining patterns within meiotic cells (*Figure 7E* and *Figure 7—figure supplement 1C*). Notably, H3K23me3 appeared relatively punctate across the entire nucleus rather than being enriched on either the autosomes (such as H3K4me3) or the silenced Xs (such as H3K27me3) (*Kelly et al., 2002*; *Bender et al., 2004*). Importantly, our peptide competitions showed that α-H3K23me3 is highly specific for H3K23me3 in the *C. elegans* gonad (*Figure 7—figure supplement 1D*).

Since an E(z) (Enhancer of zeste) homolog is required for H3K23 methylation in *Tetrahymena*, we examined whether the only E(z) homolog in *C. elegans*, the H3K27 methyltransferase MES-2, could also methylate H3K23. As expected, in the gonad of *mes-2* knockout strains, H3K27me3 was completely undetectable (*Figure 7—figure supplement 1E*). In contrast, H3K23me3 remained present in meiotic cells, and surprisingly, H3K23me3 levels within the mitotic germ cell region increased (*Figure 7—figure supplement 1E*). Additionally, we investigated H3K23me3 levels following RNAi-mediated

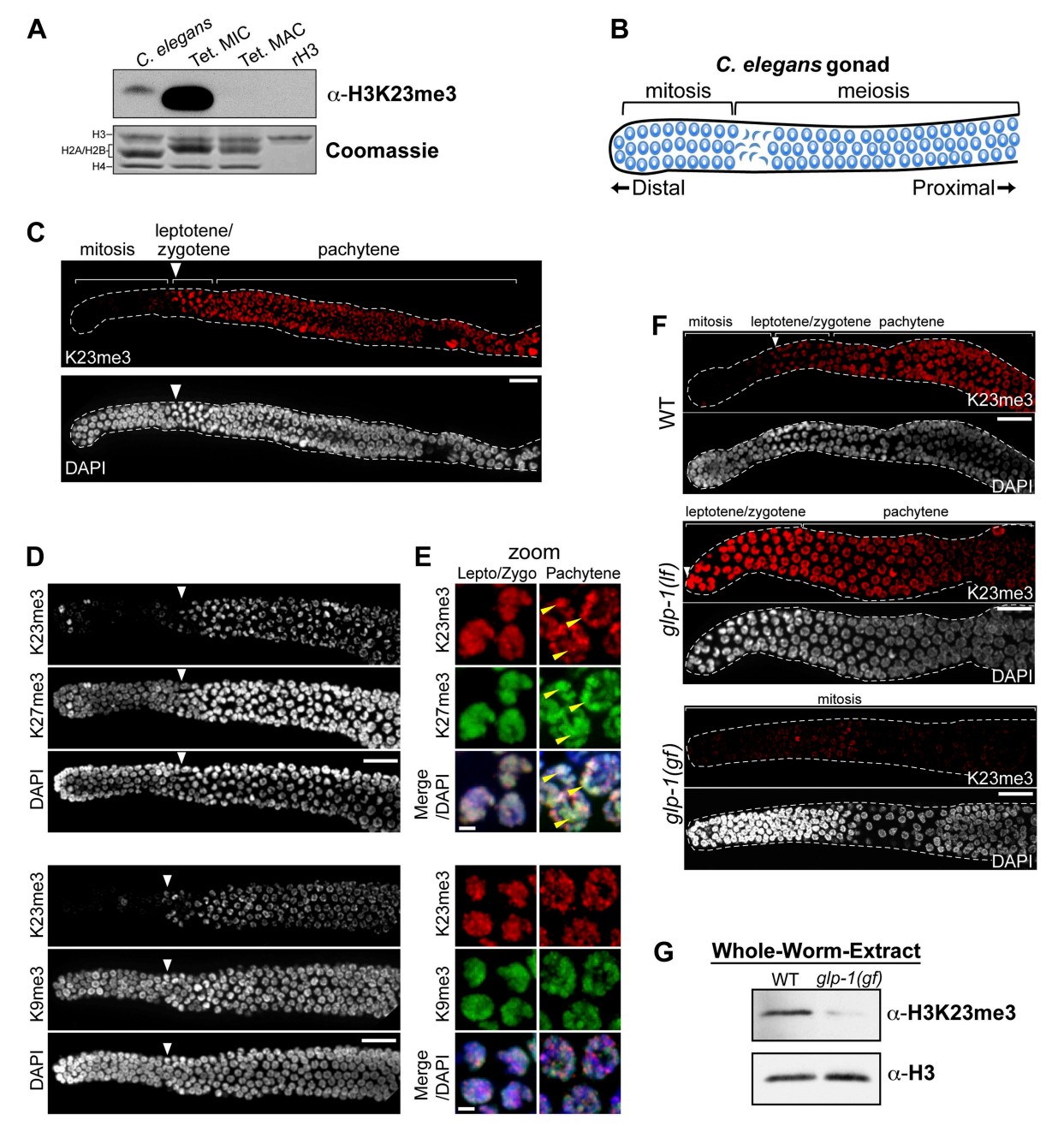

**Figure 7**. H3K23me3 is conserved in the *C. elegans* germline. (**A**) Western blot analysis of nuclear acid-extracts using α-H3K23me3. *C. elegans* embryonic nuclei, Tetrahymena micro- or macronuclei, and recombinant human histone H3 (rH3) were resolved by SDS-PAGE and normalized by Coomassie staining. (**B**) Schematic of *C. elegans* adult hermaphrodite gonad. (**C**) Immunofluorescence staining of formaldehyde fixed *C. elegans* gonad stained with α-H3K23me3 and counterstained with DAPI. Scale bar, 10 µm (**C**, **D**, and **F**). White arrowheads point to the mitotic/meiotic transition zone (**C**, **D**, and **F**). (**D**) Co-immunofluorescence staining of the *C. elegans* gonad using α-H3K23me3 and either α-H3K9me3 or α-H3K27me3. (**E**) Zoomed images of germline nuclei co-stained with antibodies targeting the indicated H3 PTMs. Yellow arrowheads point to the location of the silenced X chromosomes. Scale bar, 2 µm. (**F**) H3K23me3 localization in loss-of-function and gain-of-function mutants (*lf* or *gf*, respectively) of the Notch-like receptor GLP-1. (**G**) Western blot analysis of whole-worm-extracts to evaluate total levels of H3K23me3 in *glp-1(gf)* worms with defective meiosis. See also **Figure 7—figure supplement 1** for supporting information.

The following figure supplement is available for figure 7:

**Figure supplement 1**. Conservation of H3K23me3 enrichment during meiosis in higher eukaryotes.

knockdown of MES-4, an NSD homolog known to antagonize MES-2 in the germline (*Gaydos et al., 2012*). Similar to *mes-2* knockouts, H3K23me3 remained detectable in mitotic and meiotic regions of the *mes-4(RNAi)* germline (*Figure 7—figure supplement 1F*). MES-2 and MES-4 regulate gene expression in the germline, and their disruption may have pleiotropic effects that include alteration of pathways regulating H3K23me3 (*Gaydos et al., 2012*). Nevertheless, our data suggest neither MES-2 nor MES-4 is exclusively responsible for trimethylation of H3K23 in *C. elegans*.

We further examined the correlation between H3K23me3 and meiotic entry in *C. elegans* using mutants of the Notch-like receptor GLP-1, which maintains the proliferative state (mitotic) of germline stem cells (*Crittenden et al., 1994*; *Kimble and Crittenden, 2007*). As expected, in the GLP-1 loss-of-function mutant *glp-1(lf)*, which undergoes early differentiation of mitotic cells and early entry into meiosis, we observed a shift of the H3K23me3 signal towards the distal tip of the gonad (*Figure 7F*). Remarkably, in the GLP-1 gain-of-function mutant *glp-1(gf),* which prevents entry of mitotic germ cells into meiosis and causes their proliferation, H3K23me3 levels were significantly reduced throughout the gonad (*Figure 5F*). Moreover, Western blot analysis of *glp-1(gf)* whole-worm-extracts showed that H3K23me3 levels are also dramatically reduced, further indicating the vast majority of H3K23me3 signal in adult *C. elegans* is confined to meiotic germ cells (*Figure 7G*). Together with our previous data, these results suggest the up-regulation of H3K23me3 during meiosis is conserved between *Tetrahymena* and *C. elegans*.

## Discussion

In general, little is known about the roles of histone PTMs in governing the myriad of structural changes germline chromatin undergoes during development and differentiation. In this report, we demonstrate that a poorly understood PTM, H3K23me3, is enriched in germline chromatin of *Tetrahymena* and *C. elegans*. Furthermore, in *Tetrahymena*, we show that H3K23me3 plays a role in a previously unknown pathway maintaining genome stability during meiosis, and that mutants lacking H3K23 methylation incur a reduction in progeny viability. Taken together with the conserved upregulation of H3K23me3 in meiotic germline chromatin in *Tetrahymena* and nematodes, our data suggest a previously unappreciated pathway may help limit DSB formation and recombination in heterochromatin during meiosis.

### H3K23me3 helps regulate sites of DNA damage in the meiotic germline

Increasing evidence indicates histone PTMs can act in contrasting pathways to help confine meiotic DSBs to specific 'hotspot' regions. The location of recombination hotspots in budding yeast and mice is positively correlated with the euchromatic PTMs H3K4me3 and acetylation (*Lichten and de Massy, 2011* and references therein; *Kota and Feil, 2010*; *Yamada et al., 2013*). H3K4me3 and histone acetylation may enhance recruitment of the opportunistic, sequence-independent DSB endonuclease Spo11 to specific sites in the genome by loosening local chromatin structures (*Kim et al., 2007*). Alternatively, the heterochromatic PTM H3K9me3 is thought to play a role in inhibiting meiotic DSBs and crossover events through compaction of specific genomic regions, like pericentric heterochromatin and telomeres, thus limiting Spo11 access (*Gerton et al., 2000*; *Wu et al., 2009*). While H3K9me3 and H3K4me3 are absent altogether from *Tetrahymena* micronuclei (*Figure 1D* and *Figure 3—figure supplement 1A*) (*Strahl et al., 1999*; *Taverna et al., 2002*, *2007b*), H3K23me3 is readily detected in meiotic micronuclei and found within a distinct sub-nuclear locale than DSBs during leptotene and zygotene (*Figure 6A–D*). Furthermore, depletion of H3K23me3 induces meiotic DSBs at non-coding, centromere-proximal regions of the *Tetrahymena* germline that are occupied by H3K23me3 in wild type cells (*Figure 6B*). In analyzing *ΔSPO11* cells, which lack meiotic DSBs and γH2A.X (*Mochizuki et al., 2008*), we found that H3K23me3 levels remained at wild type levels during meiosis, supporting a role for H3K23me3 that is upstream of DSB formation (data not shown). To this end, we favor a model (*Figure 6—figure supplement 2*) in which H3K23me3 helps to organize a specialized compact heterochromatin structure, distinct from H3K9me3-associated heterochromatin, that can protect non-coding/highly repetitive regions of the genome from Spo11 activity. The mechanism in *ΔEZL3* mutants that leads to an enrichment of early meiotic DSBs at pericentric regions, rather than a global derepression of DSBs, is unclear. Notably, in *Su(var)3-9[null]* flies, which lack H3K9 methylation, DNA damage is also predominantly increased in heterochromatin (*Peng and Karpen, 2009*). ChIP-based (chromatin immunoprecipitation) efforts to further refine genomic localization of DSB in H3K23me3 *Tetrahymena* mutants were problematic (data not shown), likely due to the repetitive nature of the DNA surrounding

centromeres. Since H3K23me3 is also found in some mitotic chromatin (*Figures 1 and 7*), an intriguing link may exist between this PTM and homologous recombination events outside of meiosis.

The decrease in progeny viability for the *ΔEZL3* cells missing H3K23me3 is also similar in scope to reported infertility and increased spermatocyte apoptosis in male mice missing both copies of the H3K9me3 methyltransferases, Suv39h1/2 (*Figure 6E*) (*Peters et al., 2001*). The phenotype of *Suv39h* dn mice was largely attributed to a loss of H3K9me3 at pericentric heterochromatin causing a significant amount of genome instability in both germline and somatic cells. Similarly, the obvious shift of DNA damage to pericentric regions in *Tetrahymena* mutants lacking H3K23me3 may also underlie their decreased progeny viability. Future analysis of changes in the piRNA-like scanRNA population and genome-wide localization studies of DSBs in *ΔEZL3* mutants will further resolve the relationship between histone PTMs and genome instability.

## Combinations of histone modifications within *Tetrahymena* heterochromatin

With close to 200 sites and types of individual histone PTMs described, attention is now turning toward how multiple PTMs function together within the context of their physiological chromatin template (*Ruthenburg et al., 2007*; *Taverna et al., 2007b*). Mass spectrometry data describing the physical linkage of histone PTMs is sparse in relation to their biological context, yet such information is necessary to understand the proposed 'epigenetic/histone code' (Strahl et al., 2000). Our HILIC-MS/MS data demonstrate that H3K23me3 is located within the *Tetrahymena* germline, and primarily co-exists with H3K27me3 on the proteolytically clipped H3$^F$ isoform (*Figure 1D* and *Figure 3—figure supplement 1A*). In addition to the physical linkage of these modifications, the loss of either H3K23me3 or H3K27me3 clearly impacts the level of the other in both *Tetrahymena* and *C. elegans*. Western blotting and quantitative mass spectrometry of *Tetrahymena* histones suggested an increase in H3K27me2 in cells missing H3K23me3, while indirect immunofluorescence of *C. elegans* gonads suggested H3K23me3 increased in worms lacking H3K27me3. (*Figure 4A*, *Figure 4—figure supplements 1C and 1E*, and *Figure 7—figure supplement 1E*). However, the mislocalization of meiotic DSB in *Tetrahymena* was only observed in H3K23me3 mutants (*Figure 6B*). In this regard, the functional significance of the combination of K23 and K27 methylation within the germline remains unclear. Previously, proteolysis of H3 N-termini was observed during differentiation of mouse embryonic stem cells and also linked to H3K27me3 (*Duncan et al., 2008*), although we did not detect H3K23me3 on truncated H3 from mouse or *C. elegans* (*Figure 7A* and data not shown). Interestingly, a small proportion of H3K23me3 and H3K27me3 are also found in the presence H3S10ph, which is a well-documented PTM that marks mitotic and, to a lesser extent, meiotic chromatin (*Figure 1D* and *Figure 3—figure supplement 1*). (*Wei et al., 1998*; *Prigent and Dimitrov, 2003*). Minimally, this physical linkage suggests that some amount of H3K23me3 can be retained on chromatin throughout the process of cell division, and thus can be inherited.

Finally, we speculate that as yet unknown protein complexes specifically interact with H3K23me3 in vivo, perhaps in the presence of H3K27me3, or other distinct PTM combinations (*Ruthenburg et al., 2007*; *Taverna et al., 2007a*; *Liu et al., 2010*; *Moore et al., 2013*). Identification of such effector complexes and characterization of H3K23me3 methylases and demethylases in higher eukaryotes will undoubtedly lead to a better appreciation of H3K23me3, meiosis, and disorders linked to genome instability, like cancer and infertility.

# Materials and methods

## Cell culture and general methods

*Tetrahymena* strains CU427, CU428, and B2086, provided by the *Tetrahymena* Stock Center (http://tetrahymena.vet.cornell.edu/), were used as wild type controls and maintained as previously described (*Gorovsky et al., 1975*). Unless otherwise stated, *ΔEZL2* and *ΔEZL3* strains described in the text are germline gene knockouts, in which a portion of the wild type gene, including the methyltransferase active site, has been deleted from all somatic and germline gene copies, as previously described for *ΔEZL1* (*Liu et al., 2007*). For mating survival and progeny production tests (*Figure 6E* and *Figure 6—source data 1*), the *ΔEZL3* gene deletion was reproduced as a somatic knockout (*Bruns and Cassidy-Hanley, 2000*) in the genetic background of strains CU428 and B2086. Multiple independent lines in each background were established by cloning single cells from drug-resistant transformants. Each line

was starved, mated to each of the others and 44 single mating pairs isolated into separate drops of standard growth medium (*Hamilton and Orias, 2000*). After 3 days of growth, surviving cultures were replicated to a medium containing 6-methyl purine. The 6MP-resistance allele is found in the silent germline nucleus of CU428, but not its somatic nucleus; therefore, resistance indicates the production of true genetic progeny. Survival and progeny production were scored and compared with parallel matings of the parental strains. For testing the rescue of *ΔEZL3* methyltransferase activity (*Figure 4C*), the wild type and catalytically defective mutant (H599A) *EZL3* genes were each inserted into the vector pBS-ICY-gtw (*Motl and Chalker, 2011*), and the resulting plasmids linearized by digestion with PvuII. The digested DNAs were introduced into *ΔEZL3* cells using biolistic bombardment, according to standard procedures (*Bruns and Cassidy-Hanley, 2000*). Somatic transformants were selected by resistance to cycloheximide at 12.5 µg/ml and then cultured in medium containing 25 µg/ml cycloheximide through numerous cell fissions to allow copy number of the rescue gene to increase by phenotypic assortment.

C. elegans strains were derived from *C. elegans* Bristol strain N2 and cultured using standard techniques (*Brenner, 1974*). Strain DG2389 *glp-1 (bn18lf)* was cultured at the permissive temperature (15°C) and shifted to 25°C for 18 hr to interfere with Notch signaling and induce meiotic differentiation of germ cells. Strain GC833 glp-1(ar202gf) was grown at 25°C from L1 larvae to adulthood to continuously expose germ cells to high levels of Notch signaling. To analyze loss of the *C. elegans* Enhancer of zeste homolog, we used the *mes-2(bn11)* homozygous mutant. To analyze loss of *mes-4* function, wild type *C. elegans* were treated with *mes-4(RNAi)* from L1 larvae to adulthood by a standard feeding protocol (*Timmons and Fire, 1998*).

## Preparation of micronuclear histone H3

Nuclei were isolated following suggested conditions from published protocols (*Sweet and Allis, 2006*). Nuclear fractions were stained with DAPI and monitored by fluorescence microscopy. In all experiments, micronuclear fractions containing less than 0.1% macronuclear contamination were pooled. Then, total histones were extracted in 0.4 N $H_2SO_4$ (*Taverna et al., 2002*), and the histone isoforms were separated by reversed-phase HPLC using a C8 column (220 × 4.6 mm, Aquapore RP-300, Perkin Elmer, Waltham, MA) following previously described methods (*Shechter et al., 2007*). Purity of HPLC fractions was analyzed by SDS-PAGE and Coomassie blue staining. HPLC fractions containing micronuclear histone H3 were pooled, dried under vacuum, and their PTM states were analyzed by HILIC-MS/MS.

## Histone extraction from *C. elegans* embryo nuclei

Nuclei from embryos were isolated following a previously described method (*Maddox et al., 2007*). Once isolated, nuclei were pelleted and resuspended in 5X pellet volume 0.4 N $H_2SO_4$, and rotated for 2 hr at 4°C. Following centrifugation for 10 min at 10,000×*g*, histones in the supernatant were acid-precipitated by 100% trichloroacetic acid (TCA) at a dilution factor of 1:4 (TCA: supernatant; final 20% TCA) for 1 hr on ice. TCA precipitates were pelleted by centrifugation for 10 min at 10,000×*g*, washed 2X with ice-cold acetone, resuspended in water, and stored at −80°C.

## ETD mass spectrometry

RP-HPLC fractions containing micronuclear H3 (*Figure 1C*, fractions 56–58 containing H3^F and H3^S) were pooled, dried under vacuum, reconstituted in 100 mM ammonium acetate (pH 4.1) and digested with endoproteinase Glu-C (Roche Applied Science, Indianapolis, IN) at a ratio of 1:20 enzyme to protein for 4 hr at 37°C. Digestion was quenched by freezing the samples at −30°C.

An aliquot of the endoproteinase Glu-C digested H3 histone peptide solution (5 pmole) was dried to completion in a Savant SpeedVac and reconstituted in solvent A (as described below) prior to pressure loading onto fused silica capillary analytical column (360 µm o.d. × 75 µm i.d.) containing 10 cm of PolyCAT ATM (300 Å pore size, 3 µm diameter, Poly LC Inc., Columbia, MD) equipped with an integrated electrosprayemitter tip. Peptides were gradient eluted off of the column using a linear gradient as previously described (*Young et al., 2009*). The eluent was directly introduced into a front-end electron transfer dissociation enabled LTQ-FT (Thermo Scientific, Waltham, MA) mass spectrometer. MS[1] spectra were acquired in the high-resolution FT mass analyzer and MS[2] spectra were acquired in the linear ion trap mass analyzer. Following each high resolution MS[1] spectrum, eight data-dependent low resolution ETD MS[2] spectra were acquired. All MS[2] spectra were collected using the following instrument

parameters: one microscan, 3 m/z isolation window, default charge state of +10, monoisotopic precursor selection 'enabled'. Full automated gain control targets were set to 1e$^6$ for FTMS and 6e$^5$ for ITMS. In addition dynamic exclusion was enabled and set for a repeat count of 2, repeat duration of 30 s, exclusion list of 150 masses, and exclusion duration of 20 s. Azulene was the electron transfer reagent.

All MS/MS spectra were searched against a *Tetrahymena* histone H3 protein database using the Open Mass Spectrometry Search Algorithm (OMSSA) (*Geer et al., 2004*). All searches were completed using either the 'no enzyme' or 'Glu-C' digestion parameter. Precursor and peptide mass tolerances were set to ± 0.05 and ± 0.35, respectively. Search parameters included the following variable modifications: mono-, di-, and tri-methylation of Lys; acetylation of Lys; and phosphorylation of Ser, Thr, Tyr. OMSSA removed the reduced charge species from the ETD peak lists prior to searching the data. Results from the searches were used as a guide for data analysis. All data were interpreted manually in order to determine sequence coverage and modification sites.

## Quantitative mass spectrometry

Gel bands corresponding to histones H3, H2B, H2A, and H4 were excised, subjected to in-gel chemical labeling of lysines with d6-acetic anhydride, and subjected to in-gel trypsin digestion (*Tackett et al., 2005*). Histone peptides were analyzed by LC-MS/MS with a Thermo Velos Orbitrap coupled to a Waters nanoACQUITY LC system (*Byrum et al., 2013*). Histones and methylated lysines were identified with Mascot. All four core histones were identified; however, only histone H3 contained detectable levels of methylated lysines. Spectral counts for histone H3 peptides containing lysine methylations were compared to determine relative changes in H3 methylation between wild type and *ΔEZL3* cells (*Figure 4—figure supplement 1F*).

## Generation of polyclonal antibody

α-H3K23me3 antibodies were generated by injecting rabbits (Cocalico Biologicals Inc, Reamstown, PA) with a synthetic peptide spanning amino acids 18–36 of histone H3 and containing a trimethylated lysine 23 modification (H3K23me3[18–36]).

H3K23me3-specific antibodies were affinity purified from rabbit serum as previously described (*Harlow and Lane, 1988*). Briefly, IgG from the serum was precipitated in 50% saturated ammonium sulfate and resuspended in 1× PBS +1 mM PMSF. The IgG was then passed through a gravity column containing SulfoLink beads (Thermo Scientific) coupled to the non-native cysteine of a short synthetic peptide, K23me3 (19–28), from the H3 N-terminus containing the H3K23me3 modification. Bound antibodies were eluted with 100 mM glycine pH 2.5, and the eluate was immediately neutralized to pH ~7 with 1 M Tris pH 8.0, flash frozen and stored in −80°C.

Specificity of the purified IgG was evaluated by ELISA. Greiner high-binding 96-well plates (Sigma, St. Louis, MO) were incubated with peptides diluted in 0.05 M carbonate buffer pH 9.6 overnight, blocked with 1% BSA resuspended in PBS-T (1× PBS pH 7.4 + 0.05% Tween 20) for 1 hr, incubated with affinity purified α-H3K23me3 diluted in PBS-T (1:100) for 1.5 hr, and incubated with HRP-conjugated α-rabbit secondary antibody (GE Life Sciences) diluted in PBS-T (1:5000) for 1.5 hr. All incubation steps were carried out at 37°C followed by 3× washes with PBS-T. To detect the ELISA signal, plates were treated with stable peroxide substrate buffer (Thermo Scientific) supplemented with OPD (Thermo Scientific) following manufacture's procedures, and the absorbance at 492 nm was measured using a microplate reader. Peptides were bound in triplicate in each plate, and all ELISA experiments were repeated at least three times. Representative results from a single experiment are presented, and the error bars denote standard deviation of triplicate samples. Peptide competition experiments for ELISAs were carried out by incubating the indicated peptides at a concentration of 5 μg/ml (of PBS) with the antibodies for 1 hr on ice prior to their use.

## Western blot and immunofluorescence analyses

SDS-PAGE was performed as described (*Laemmli, 1970*) or materials and procedures corresponding to Novex NuPAGE precast gel system were used (Life Technologies, Grand Island, NY). The procedures for methanol fixation and indirect immunofluorescence staining of *Tetrahymena* cells are described previously (*Madireddi et al., 1994*).

Rabbit polyclonal antibodies used in Western blots included: α-H3 (1:5000; gift from CD Allis), α-H3K23me3 (1:100; made in this study), α-H3K23me3 (1/1000; Active Motif, Carlsbad, CA; used in *Figure 4C*), α-H3S10p (1:25,000; gift from CD Allis), α-H3K9me3 (1:1000; ab8898; Abcam, Cambridge,

England), α-H3K27me3 (1:2500; 39155; Active Motif), α-alpha linker (1:5000; gift from CD Allis), α-H4 penta acetyl (1:5000; gift from CD Allis), α-Pdd1p (1:5000; ab5338; Abcam), α- H3K23me1 (1:1000; 39387; Active Motif), and α-H3K23me2 (1:1000; 39653; Active Motif). HRP-conjugated α-rabbit secondary antibodies (GE Life Sciences, Pittsburgh, PA) were used at dilution of 1:5000 and detected by ECL plus reagent (Amersham, Pittsburgh, PA) following manufacturer's procedures.

Tetrahymena indirect immunofluorescence staining: cells were fixed in partial Schaudin's fixative as described (Madireddi et al., 1994) (Wenkert and Allis, 1984). The primary antibodies used were: rabbit α-H3K27me3 (1:250; gift from T Jenuwein), mouse α-H3K4me3 (1:250; ab1012; Abcam), rabbit α-H3K23me3 (1:100), rabbit α-H2A (1:250; control strain for Tetrahymena chromatin), mouse α-H3K9me3 (1:100; 39285; Active Motif), mouse α-γH2AX (1:250; 613402; BioLegend), rabbit α-Cna1p (1:5000; gift from HS Malik), rabbit α-H3K23me1 (1:250; Active Motif), and rabbit α-H3K23me2 (1:250; Active Motif). Secondary antibodies, used at a dilution of 1:2000, included Alex Fluor 647-conjugated goat α-mouse, Alexa Fluor 488-conjugated goat α-rabbit, and Alex Fluor 488-conjugated goat α-mouse (Life Technologies). Slides were mounted in ProLong Gold antifade reagent with DAPI (P36931; Life Technologies) and stored overnight at room temperature prior to imaging. Digital images were captured using a Zeiss Axio Observer fluorescence microscope equipped with an Apotome for optical sectioning and AxioCam MRm camera. Raw images were exported as tiff files and processed using Adobe Photoshop software CS3.

Fluorescence in situ hybridization coupled with immunofluorescence: mating Tetrahymena cells conjugating between 2 and 4 hr were fixed in partial Schaudin's fixative as described (Madireddi et al., 1994) (Wenkert and Allis, 1984), and dropped on slides. DNA-immunoFISH was adapted from Reddy et al., 2008. Briefly, slides were rehydrated in PBS for 20 min, and fixed again in 1% paraformaldehyde/PBS solution for 5 min, and washed three times with PBS. Slides were then RNAse treated (100 µg/ml) for 15 min in 2× SSC at room temperature in a humidified chamber. Slides were hybridized by washing in 70% formamide/2x SSC at 76°C for 3 min, then 50% formamide/2x SSC at 76°C for 1 min, and then incubating with denatured and preannealed probe/hybridization solution (Cy5 labeled Tlr1 probe, 9 µg salmon sperm DNA, 6 µg placental DNA, 10% dextran sulfate, 50% formamide, and 2× SSC) overnight at 37°C in a humidified chamber. Probe to Tlr1 element was generated by directly labeling Tlr1 PCR product with Cy5 (Primers: Forward - TCGGATCGGATTGGATTAAC, Reverse -CCTGGAGCTTACCGTCTTTG; Aglient Genomic DNA ULS labeling kit 5190-0419). After overnight incubation, slides were washed three times in 50% formamide/2× SSC at 47°C, three times with 63°C 0.2× SSC, one time with 2× SSC, and then two times with PBS before blocking with 4% BSA in PBS for 30 min in a humidified chamber. Slides were then incubated with primary antibody (α-γH2AX [1:100; 613402; BioLegend, San Diego, CA]) in blocking medium overnight at 4°C. Slides were washed three times with PBS/0.05% Triton X-100 and then incubated with secondary antibody a in blocking medium (1:100; DyLight 488, Jackson ImmunoResearch, West Grove, PA) for 1 hr at room temperature. Post incubation, slides were washed three times with PBS + 0.05% Triton X-100, and then DNA counterstained with 1 µg/ml Hoechst. Slides were then washed, mounted with SlowFade Gold (Life Technologies), after which images were collected and processed as described above.

Worm immunofluorescence staining: adult C. elegans hermaphrodites were washed in PBS, and germlines were dissected on poly-L-lysine treated slides, covered with a coverslip to ensure attachment to slide, and snap-frozen on aluminum blocks on dry ice. The samples were fixed for 1 min in 100% methanol at −20°C, followed by 5 min in 2% EM grade paraformaldehyde in 100 mM $K_2HPO_4$ pH 7.2 at room temperature. The samples were blocked for at least 30 min with PBS containing 0.1% BSA and 0.1% Tween-20 (PBTw) with 10% Normal Goat Serum. Primary antibodies were diluted in PBSTw as follows: rabbit α-H3K23me3 (1:70); mouse α-H3K9me3 (1:100; 39285; Active Motif); mouse α-H3K27me3 (1:100; 39536; Active Motif); mouse α-H3K4me3 (1:250; ab1012; Abcam). Secondary antibodies were goat α-rabbit or α-mouse conjugated with Cy3 or Alexa-488 diluted at 1:200 in PBTw (Jackson Immunoresearch and Invitrogen). Primary antibody incubations were kept overnight at 4°C; secondary antibody incubations were kept for 2 hr at room temperature. For peptide competition experiments, α-H3K23me3 was incubated with 5 µg/ml of indicated peptides for 1 hr on ice, and immune complexes were removed by centrifugation for 10 min at 14,000 rpm. Preabsorbed primary antibodies were applied to the tissue samples for 1 hr at room temperature followed by secondary antibody incubation for 1 hr at room temperature. Before imaging, DNA was counterstained with DAPI.

Microscopy for worm slides: confocal images were acquired with a Cascade QuantEM 512SC camera (Photometrics, Tucson, AZ) attached to a Zeiss Axioimager microscope with Yokogawa spinning

disk confocal scanner and Slidebook software (Intelligent Imaging Innovations, Denver, CO). Maximal-intensity projections encompassing a single layer of nuclei were generated by the Slidebook software. Image processing was performed in Adobe Photoshop CS4.

## Observation of small RNAs in *Tetrahymena*

Total RNA was extracted with Trizol reagent (Life Technologies) according to manufacturer's procedures and precipitated in ethanol. RNA pellet was dissolved in water and the $OD_{260nm}$ was measured. Total RNA (15 µg) from conjugating cells was diluted in 2× loading buffer (95% formamide, 18 mM EDTA, 0.025% SDS, and Bromophenol Blue) and resolved in a denaturing acrylamide gel (1X TBE, 7 M Urea, 15% acrylamide) according to previously described methods (*Liu et al., 2007*; 'Gel purification of miRNA from total RNA' protocol from Life Technologies). Gels were visualized by soaking in 1.5 µg ethidium bromide per ml of TAE buffer (*Mochizuki et al., 2002*).

## Peptide synthesis

Peptides were synthesized on a Prelude Synthesizer (Protein Technologies, Tucson, AZ) using standard Fmoc-based solid phase peptide chemistry. Peptides were cleaved from the resin by Reagent K, precipitated in diethyl ether, and purified using Varian Dynamax Microsorb C18 preparative column (Agilent, Santa Clara, CA). The masses of HPLC-purified peptides were confirmed by MALDI-TOF mass spectrometry.

## Histone H3 Peptide sequences

Unmod (18–36): KQLASK$_{23}$AARK$_{27}$SAPATGGIK(biotin)
K23me3 (18–36): KQLASK$_{23}$(me3)AARK$_{27}$SAPATGGIK(biotin)
K23me3 (19–28): QLASK$_{23}$(me3)AARK$_{27}$C
K27me3 (18–36): KQLASK$_{23}$AARK$_{27}$(me3)SAPATGGIK(biotin)
K23/K27me3 (18–36): KQLASK$_{23}$(me3)AARK$_{27}$(me3)SAPATGGIK(biotin)
Unmod (1–20): ARTK$_4$QTARK$_9$STGGKAPRKQLY
K4me3 (1–15): ARTK$_4$(me3)QTARK$_9$STGGKAY
K9me3 (1–15): ARTK$_4$QTARK$_9$(me3)STGGKAY
K23me1 (20–28): CLATK$_{23}$(me1)AARK$_{27}$S
K23me2 (20–28): CLATK$_{23}$(me2)AARK$_{27}$S

## PCR, cloning the EZL3 gene, and mutagenesis

To check the expression of *EZL3* in *EZL* knockout strains, indicated strains were grown to mid-log phase, their total RNA was extracted with Trizol reagent (Life Technologies) and precipitated in ethanol. Samples were resuspended in water and treated with TURBO DNase (Life Technologies) following the manufacturer's procedures. Reverse strand synthesis was carried out using poly dT primers, procedures and reagents from the Superscript II RT kit (Life Technologies). *EZL3* specific primers (Forward -AATGGTCTATGCTAAAGGTGG; Reverse -TTTGAGAGTTCCTTGTCCATC) were used to assess transcript levels and primers that target the ribosomal protein RPL21 (Forward - AAGTTGGTTATCAACTGTTGCGTT and Reverse -GGGTCTTTCAAGGACGACGTA; sequence obtained from *Aronica et al., 2008*) were used as controls in end-point PCR assays. Quantitative PCR was employed to assess the levels of *EZL3* expression during conjugation. Total RNA of conjugating cells was harvested at the indicated time points during conjugation and processed following procedures described above. qPCR was performed using an Applied Biosystems StepOne Plus Real-Time PCR System (Life Technologies, 4376600) with StepOne Software v2.1 and Power SYBR Green (Life Technologies, 4367659) to quantify relative mRNA abundance. *EZL3* specific primers used are shown above. mRNA levels were normalized to time point zero. Samples were analyzed in duplicate. The *EZL3* gene was amplified from micronuclear genomic DNA using the forward primer (5′ -CACCTATAAAATGAGAA GAGTTAGATAAAATAAATG) and the reverse primer (5′ -GATATCTTTCCTTTTTTTTTATTTTATTAAA TTTTAATTTTTAGCAG) and cloned into the pENTR-TOPOD (Life Technologies). This plasmid was used as the template for the site directed mutagenesis (Stratagene, La Jolla, CA) reaction that introduced the H599A mutation. The primers for mutagenesis were: forward primer (5′ -GCAGATTCAT GAATGCTTCAAAAAGCAAAG) and reverse primer (5′ -CTTTGCTTTTTGAAGCATTCATGAATCTGC). The resultant plasmids (pENTREZL3wt and pENTR-EZL3[H599A]) were then cloned into the pBS-ICY-gtw destination vector using the LR recombination reaction as described previously (*Motl and Chalker, 2011*). All plasmids and inserted genes were confirmed by sequencing.

## Acknowledgements

We thank CD Allis for providing several *Tetrahymena* general histone antibodies and helpful discussions, PA Cole, W Timp, X Chen for critical reading of this manuscript, BA Garcia for assistance with HILIC-MS/MS, H. Malik for α-Cna1, MK Tarrant for assistance with peptide synthesis, G Seydoux for critical review of *C. elegans* data and helpful suggestions, J Loidl for providing the *SPO11* knockout cells, DL Chalker for providing the pBS-ICY-gtw expression plasmid, A. Raman for assistance with cloning, TN Mavrich for help with generating *Tetrahymena* strains and A Schuller, B Barrera, E Clay, and other members of the Taverna laboratory for their support. *C. elegans* strain DG2389 was provided by *Caenorhabditis* Genetics Center funded by the NIH. We would like to acknowledge the UAMS Proteomics Facility for mass spectrometric support. We thank the following for grant support: a [DBP] award to SDT from NIH Grant U54 RR020839 (Awarded to J Boeke and JHU), NSF grant DGE-0707427 to TG, and NIH grants 5F32GM080923 to EV, GM089662 to RSC, GM37537 to DFH, GM087343 to YL, R01GM106024 to SDT and AJT, P30GM103450 to F Millett, P20GM103429 to L Cornett, and UL1TR000039 to L James.

## Additional information

### Funding

| Funder | Grant reference number | Author |
| --- | --- | --- |
| National Institute of General Medical Sciences | R01GM106024 | Alan J Tackett, Sean D Taverna |
| National Science Foundation | DGE-0707427 | Tonya M Gilbert |
| National Institutes of Health | P30GM103450 | Alan J Tackett |
| National Institutes of Health | P20GM103429 | Alan J Tackett |
| National Institutes of Health | UL1TR000039 | Alan J Tackett |
| National Institutes of Health | U54RR020839 | Sean D Taverna |
| National Institutes of Health | GM089662 | Robert S Coyne |
| National Institute of General Medical Sciences | 5F32GM080923 | Ekaterina Voronina |
| National Institute of General Medical Sciences | GM37537 | Donald F Hunt |
| National Institutes of Health | GM087343 | Yifan Liu |

The funders had no role in study design, data collection and interpretation, or the decision to submit the work for publication.

### Author contributions

RP, EV, JRC, KLR, SDT, Conception and design, Acquisition of data, Analysis and interpretation of data, Drafting or revising the article; TRL, JS, AJT, DFH, Conception and design, Acquisition of data, Analysis and interpretation of data; TMG, SGM, Acquisition of data, Analysis and interpretation of data; EM, Acquisition of data, Drafting or revising the article; RSC, Conception and design, Acquisition of data, Analysis and interpretation of data, Drafting or revising the article, Contributed unpublished essential data or reagents; YL, Conception and design, Contributed unpublished essential data or reagents

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
