## [Decision Letter]

Your manuscript titled “Methylation of histone H3K23 blocks DNA damage in pericentric heterochromatin during meiosis” was reviewed by two experts in the field and by a member of the Board of Reviewing Editors (BRE). After a full discussion of the study and the reviews, I am happy to report that the reviewers and the BRE member found the study of great interest to the journal and therefore we are happy to consider a revised manuscript addressing the following issues.

1) The EZL enzymes need to be expressed, purified, and their methylase activities toward H3K23 and H3K27 on either H3 or nucleosomes with K23R or A as internal controls need to be demonstrated.

2) In the revised study, you and co-authors should present a statistical analysis of the results. Are they significantly different as they stand? More importantly, you and co-authors should prepare some additional wild-type and mutant strains and add those to the analysis. In *Tetrahymena*, it should be very straightforward to generate a set of mutants to compare with the wild-type.

3) It is also very important to demonstrate in the revised manuscript more directly that there are new DSBs in the pericentric heterochromatin.

4) The mouse data, and to some degree the *C. elegans* data, are definitely out of place. Please either further develop this data per the reviewers' suggestions (as included below) or just cut this portion out of the paper and just go with the *Tetrahymena* portion of the study.

---

## [Author Response]

*1) The EZL enzymes need to be expressed, purified, and their methylase activities toward H3K23 and H3K27 on either H3 or nucleosomes with K23R or A as internal controls need to be demonstrated*.

Respectfully, we believe that that these experiments, while desirable, are not essential to the conclusions of the paper. The activity of a methyltransferase *in vitro* does not unambiguously define enzyme lysine preference *in vivo*. *In vitro* site specificity for methyltransferases (as is the case with most other chromatin-associated enzymes) can vary depending on the substrate, reaction conditions, the stability of the protein, and the associated complex members present in the assay. The EZH2 class of methyltransferases (and by extension, the homologous *Tetrahymena* EZL methyltransferases) can be particularly sensitive to *in vitro* assay conditions since other PRC2 complex members are essential for activity. Despite these limitations of the *in vitro* biochemistry approaches, we have spent over two years attempting to purify the *Tetrahymena* Ezl3 methyltransferase and complex using *Tetrahymena* cells, as well as heterologous cell systems.

Recombinant Ezl3 does not purify in a soluble form, and while we have previously tagged and purified endogenous Ezl1 complex ([41], and data not shown), our efforts to similarly tag endogenous Ezl3 have failed to produce viable strains. In the revised manuscript we have tried to soften language throughout regarding direct modification of H3K23 by Ezl3. Since our exhaustive efforts to characterize Ezl3 *in vitro* have been hindered by insurmountable technical barriers that will likely take many additional months or years to overcome, and given the scope of this manuscript, we ask that the reviewers provide some leeway in their suggested biochemical experiments. We also point out that EZH methyltransferases are well-documented histone methyltransferases, and ascribing Ezl3 as the key enzyme responsible for H3K23 methylation is the Occam's razor model for our data. Moreover, as knockdown of Ezl3 does not change any other histone methylation site, there is no evidence that Ezl3 causes an indirect, cascade effect that leads to H3K23 methylation. Accordingly, we hope it will not be viewed as a major detraction to exclude this data.

*2) In the revised study, you and co-authors should present a statistical analysis of the results. Are they significantly different as they stand? More importantly, you and co-authors should prepare some additional wild-type and mutant strains and add those to the analysis. In* Tetrahymena*, it should be very straightforward to generate a set of mutants to compare with the wildtype*.

For our revised manuscript, we generated and mated a larger group of *Tetrahymena* strains, and performed statistical analysis on our progeny viability results. This data (in revised Figure 6) supports our original conclusion that progeny viability is indeed significantly reduced in cells missing H3K23me (p-value < 0.005). The revised figure includes 10 new wild type crosses, and 6 newly generated and crossed mutant strains. We thank the Reviewers for these suggestions as they further support an important point of this manuscript.

*3) It is also very important to demonstrate in the revised manuscript more directly that there are new DSBs in the pericentric heterochromatin*.

We were able to develop an assay combining FISH and IF (immunoFISH) on *Tetrahymena* (as suggested by the Reviewers) to provide additional evidence that meiotic DSBs become enriched in pericentric regions in cells missing H3K23me3, and we include this new data in the revised Figure 6. A FISH probe for the germline-limited highly-repetitive Tlr1 sequence, which is present in hundreds of copies at pericentric regions in the micronucleus, has a pattern of strong overlap with γH2AX signal in cells that are missing H3K23me3, in contrast to the non-overlapping pattern of these markers in wild type cells. We have also moved the model figure to Figure 6–figure supplement 3, as unambiguous validation of this molecular model is not experimentally possible at this time.

*4) The mouse data, and to some degree the* C. elegans *data, are definitely out of place. Please either further develop this data per the reviewers' suggestions (as included below) or just cut this portion out of the paper and just go with the* Tetrahymena *portion of the study.*

We agree with the Reviewers that the mouse data is not as developed as our *Tetrahymena* and *C. elegans* data. In the original draft, we had analyzed mouse germ tissue to strengthen the link between H3K23me3 and pericentric or otherwise heterochromatic DNA regions that we observed in *Tetrahymena*, since these sequences are not well characterized in the ciliate model. We feel the new immunoFISH analysis of meiotic *Tetrahymena* chromatin that we include in our revised submission better resolves the localization of pericentric regions, and we will remove the mouse data from this submission as per the Reviewers’ request (see revised Figure 7). However, we feel that the *C. elegans* data remains valuable to maintain in this manuscript since the multicellularity and morphology of *C. elegans*, and our analysis of the available meiotic mutants, all provide critical support of our central claim that the majority of H3K23me3 is confined to meiotic chromatin. The *C. elegans* data also underscores that biological roles of H3K23me3 are likely to be distinct from other heterochromatic histone PTMs like H3K27me3 and H3K9me3. These major points would be less convincing if our data was limited to the single-celled *Tetrahymena* model. Also, we were not attempting to suggest that our data proves that H3K23me impacts *C. elegans* DSB localization, and the confusing language has been removed in our revised draft.